# GM1 Is Cytoprotective in GPR37-Expressing Cells and Downregulates Signaling

**DOI:** 10.3390/ijms222312859

**Published:** 2021-11-27

**Authors:** Ellen Hertz, Marcus Saarinen, Per Svenningsson

**Affiliations:** Department of Clinical Neuroscience, Karolinska Institutet, Biomedicum, J5:20, Karolinska University Hospital, Solna, 171 64 Stockholm, Sweden; marcus.saarinen@ki.se

**Keywords:** GPR37, GM1, Parkinson’s disease, GPCR, orphan receptor, cytoprotection

## Abstract

G-protein-coupled receptors (GPCRs) are commonly pharmacologically modulated due to their ability to translate extracellular events to intracellular changes. Previously, studies have mostly focused on protein–protein interactions, but the focus has now expanded also to protein–lipid connections. GM1, a brain-expressed ganglioside known for neuroprotective effects, and GPR37, an orphan GPCR often reported as a potential drug target for diseases in the central nervous system, have been shown to form a complex. In this study, we looked into the functional effects. Endogenous GM1 was downregulated when stably overexpressing GPR37 in N2a cells (N2a^GPR37-eGFP^). However, exogenous GM1 specifically rescued N2a^GPR37-eGFP^ from toxicity induced by the neurotoxin MPP+. The treatment did not alter transcription levels of GPR37 or the enzyme responsible for GM1 production, both potential mechanisms for the effect. However, GM1 treatment inhibited cAMP-dependent signaling from GPR37, here reported as potentially consecutively active, possibly contributing to the protective effects. We propose an interplay between GPR37 and GM1 as one of the many cytoprotective effects reported for GM1.

## 1. Introduction

To date, 34% of FDA-approved drugs target G-protein-coupled receptors (GPCRs) [1]. GPCRs are 7-transmembrane receptors embedded in the cellular lipid membrane. This allows for interactions with functionally active membrane lipids, e.g., gangliosides, which are silic acid-containing glycosphingolipids that are highly expressed in the central nervous system (CNS). Gangliosides are today known to be involved in vastly different processes including development, neuronal differentiation, and modulating receptor signaling [2]. The heterogeneity partly derives from the modifiable oligosaccharide chain, which enables specific interactions with proteins [3]. GM1, the structurally simplest of the gangliosides, accounts for 28% of the brain’s gangliosides [4] and is highly enriched in mammal myelin [5]. GM1 specifically facilitates protein folding and is a key constituent of lipid rafts, the plasma membrane microdomains that orchestrate cellular signaling [6]. Hence, GM1 has both distinct binding partners and the possibility to modulate a number of neuronal functions by regulating the assembly of signaling complexes, e.g., GPCRs [7].

Apart from a substantial role in membrane organization, GM1 has gained clinical attention as a neuroprotective agent. In addition, several neurodegenerative diseases have been linked to reduced ganglioside levels, including Parkinson’s disease (PD). B4GALNT1 knockout mice, an enzyme upstream of GM1 production, develops parkinsonism [8]. In the same model LIGA 20, a GM1 derivative, reduced α-synuclein accumulation, which is the pathological hallmark of PD [8]. In PD patients, several of the major gangliosides are reduced, particularly in the substantia nigra (SN), which degenerate in PD [9,10]. In line with this, reports have shown decreased GM1-producing enzymes in SN, explaining one pathway for reduced GM1 levels [11]. In two randomized controlled trials, GM1 treatment both reduced symptoms during treatment and reduced disease progression, indicating a disease-modifying effect [12,13]. However, the mechanisms underpinning neuroprotection are only partially explained in the literature. GM1 interacts with GFRα and RET, receptors for the neurotrophic agent glial cell-derived neurotrophic factor (GDNF), which is essential for neuronal long-term survival, and disrupted GDNF signaling is therefore proposed as a mechanism [14]. In addition, using fluorescence cross-correlation spectroscopy, researchers showed GM1 to diffuse together with G-protein-coupled receptor 37 (GPR37), indicative of a functional interaction [15].

GPR37 is a class A orphan GPCR with a long N-terminus and with highest sequence similarity to the endothelin and bombesin receptors [16,17,18,19]. It has gained much attention due to its link to parkinsonism. In the early 2000s, increased insoluble GPR37 was reported in autosomal recessive juvenile parkinsonism [20], and staining for GPR37 was claimed to be detected in Lewy bodies, the α-synuclein containing aggregates in PD brains [21]. On the basis of its localization, GPR37 was also shown to be protective against neurotoxins, e.g., MPP+ in differentiated N2a cells [22]. However, little is known about the physiological function of GPR37. Recent evidence suggests that it serves as a negative regulator of myelination [23]. Interestingly, both in the central and peripheral nervous systems, myelination is linked to adhesion GPCRs known for their large, usually cleavable, N-terminal domains [24], similar to the structure and of GPR37 [25]. GPR37 is expressed in oligodendrocytes [26] in addition to selective neuronal expression [20,27]. A number of molecules have been suggested as endogenous ligands [28,29,30] including a neuroprotective protein, prosaposin [31,32], although this remains to be confirmed [18]. The active peptide of prosaposin, TX14(A), was shown to decrease cAMP levels in astrocytes [31,32] consistent with GPR37 being Gα_i/o_ coupled. Interestingly, prosaposin, previously known as a sphingolipid activator, has been shown to bind and transport GM1 within the cell [33]. In addition, the interaction between GM1 and GPR37 was also partly prosaposin-dependent, suggesting a functional complex of the three molecules [15].

In the present paper, we hypothesized that GPR37 may mediate part of the cytoprotective effect of GM1 and performed a series of experiments to test this hypothesis and to clarify the cellular mechanism.

## 2. Results

### 2.1. N2a^GPR37-eGFP^ Cells Expressed Decreased Levels of GM1

Since GPR37 was shown to colocalize with GM1 in lipid rafts, levels of total GM1 were initially assessed using dot blot. Stable N2a^eGFP^ and N2a^GPR37-eGFP^ cells, generated in parallel, were lysed, and dot blot showed a sharp decrease of total GM1 in N2a^GPR37-eGFP^ cells (Figure 1A). GM1 is located both in plasma membrane and in intracellular compartments [34], potentially with different features. Therefore, live cells were labeled with Cholera Toxin subunit B (CtxB) before fixation according to protocol from Maglione et al. [35] to mark only the surface plasma membrane fraction of GM1. Fluorescent imaging showed a decrease of GM1 labeling at the plasma membrane in N2a^GPR37-eGFP^ compared to N2a^eGFP^ cells (Figure 1B). The detected decrease is similar in ratio to that seen in the total GM1 measurement, indicating that the membrane fraction is primarily affected. In transiently transfected N2a, the GM1 levels did not decrease after 48 h (data not shown).

### 2.2. N2a^GPR37-eGFP^ Cells Were Selectively Protected by GM1 Treatment

As GM1 is known for its neuroprotective effect, stable N2a^eGFP^ and N2a^GPR37-eGFP^ cells were sequentially treated with GM1 and MPP+. Twenty-four hours after MPP+ treatment, cell survival was evaluated with rezasurin. N2a^GPR37-eGFP^ cells were specifically rescued with GM1 treatment (Figure 2A). This effect was seen at several concentrations of MPP+ (Appendix A). In order to investigate whether the GM1 effect is GPR37-specific, we transiently transfected N2a^WT^ with plasmids for eGFP, GPR37-eGFP, or dopamine receptor 2 (D2R)-TdTomato, another Gα_i/o_-coupled receptor. In transiently transfected GPR37-eGFP cells, preincubation with GM1 was protective against MPP+ treatment (Figure 2B). The same trend was seen at several concentrations of MPP+ and time points. Intermittently increased survival with GM1 treatment was also detected with eGFP and D2R-transfected cells; however, GPR37 always showed the largest protective effect.

### 2.3. GM1 Decreased GPR37 Signaling

HEK293 is one of the most common cell line for heterologous expression of GPCRs [36]. For measuring cAMP, HEK293T was transiently transfected with eGFP, GPR37-eGFP, or D2R-TdTomato. At baseline, GPR37-eGFP cells had lower levels of endogenous cAMP than eGFP-expressing cells (Figure 3A). This indicates Gα_i/o_-coupling of GPR37 with either a ligand in the media continuously stimulating the receptor or constituent activity as previously proposed [18]. GM1 treatment for 2 h during cAMP build-up diminished the difference, returning cAMP levels back to eGFP at baseline (Figure 3A). As a positive control, cAMP levels in D2R-transfected cells were measured and responded robustly to quinpirole treatment. No effect was seen on D2R signaling, with GM1 treatment showing specificity of a GPR37 interaction (Appendix A). 

### 2.4. GPR37 Expression Level Was Not Changed with GM1 Treatment

GPR37 has previously been reported to induce cytoprotective effects when overexpressed [22]. Therefore, we investigated if GM1 treatment altered GPR37 expression with qPCR. However, we did not see changes on the transcription level (Figure 3B). In addition, the enzyme responsible for GM1 production (B3GALT4) was not changed following GM1 treatment. Surprisingly, even though N2a^GPR37-eGFP^ had marked decreased levels of GM1, B3GALT4 expression remained indifferent, indicating another pathway of GM1 reduction (Figure 3C).

## 3. Discussion

Neuroprotective agents, as well as their mechanisms of action, are constantly in focus due to the clinical potential. Hence, both GM1 and GPR37 have gained much attention the last decades. The functional complex, detected by fluorescence correlation spectroscopy, between the two is therefore intriguing [15]. It is possible that due to limited membrane permeability, previous studies have intermittently reported the absence of a neuroprotective effect of GM1 [37]. Exogenous GM1 does not pass the surface plasma membrane, which could explain why we did not detect a protective effect in N2a^eGFP^ cells. In our model, GM1 was only able to modulate GPR37 at the plasma membrane and likely exerted cytoprotective effects via membrane-specific actions. However, due to the diversity of actions linked to GM1, from general effect on lipid raft assemblage to direct protein interactions, the molecular mechanism of GM1-mediated cytoprotection is likely linked to multiple different processes.

GPR37 has been reported to exert both toxic and protective actions, depending on cellular model [20,22,31]. The physiological role as well as the signaling cascade of GPR37 is still under debate. Here, we report decreased cAMP levels in HEK293T cells supporting Gα_i/o_-coupling of GPR37. HEK293T cells are known to express relatively few endogenous receptors, but they do express GPR37 [36]. Hence, there is a possibility that the cells also secrete a ligand that continually activates GPR37. The receptor might also be constituently active, an effect seen in other N-terminally cleaved GPCR [38,39]. The increase of cAMP with GM1 treatment, indicating less GPR37 activity, could be due to either reorganization of lipid rafts that influence signaling, inhibition of ligand binding, or direct interaction between GPR37 and GM1. Interestingly, a sphingolipid binding domain (SBD) in an extracellular loop in serotonin receptor 1A (5-HT_1A_) had high similarity with an extracellular loop of GPR37 ([40,41], also see the Appendix A section). Specifically, GM1 has been shown to bind the SBD in 5-HT_1A_, with the highest occupancy in a lysine and tryptophan residues, both conserved in GPR37 [41]. The possibility of a direct binding motif should be further investigated as it would establish the precise mechanism whereby GM1 is modulating GPR37 signaling.

In our stable cell line, N2a^GPR37-eGFP^, GM1 level was decreased. However, qPCR revealed variable but on average similar levels of B3GALT4, known to be decreased in PD patients. Since the variation was similar in both cell types, this might reflect transcriptional changes, and the lower GM1 levels could be explained either by increased degradation of GM1 in N2a^GPR37-eGFP^ or decreased enzyme levels upstream of B3GALT4, such as B4GALNT1, which has been used as a model of GM1-induced parkinsonism. In addition, in our GPR37-dependent model, the reduced levels of GM1 were not the primary reason for the rescue effect since the same effect was seen in transiently transfected cells where GM1 levels were unchanged. However, in transient transfected cells, the cytoprotective effect was detected at 48 h and with 50 µM GM1. Preliminary data indicated a tendency towards toxicity with higher concentrations. This time and dose discrepancy between transient and stably overexpressing cell lines might reflect the cellular stress induced by transfection or, for the concentration, be attributable to the reduced endogenous levels of GM1 in stable N2a^GPR37-eGFP^. While GM1 is mostly known for cytoprotective actions, increased levels of GM1 leads to neurodegeneration clinically [42]. Furthermore, as seen by qPCR, GPR37 expression was unaltered with GM1 treatment, indicating that the effects seen on signaling, and not expression levels, were responsible for the effect.

## 4. Materials and Methods

### 4.1. Cellculture

To generate stable cell lines, we transfected wild-type N2a cells (N2a^WT^) from ATCC using Lipofectamine 2000 with either eGFP- or GPR37-eGFP-expressing plasmids to generate N2a^eGFP^ and N2a^GPR37-eGFP^, respectively. Twenty-four hours post-transfection, selection with 500 µg/mL G418 started, and after control cells died, single clones were selected. Both HEK293T and N2a cells were propagated in DMEM with 10% FBS, 1% penicillin–streptomycin, 1x NEAA, 1x GlutaMAX, 1 mM HEPES, and 1x sodium pyruvate. All products for cell culture were purchased from ThermoFisher (Waltham, USA). Imaging confirming expression and trafficking of GPR37-eGFP construct can be found in [43]. qPCR experiments showed an average of 100-fold increase of GPR37 transcripts in N2a^GPR37-eGFP^ compared to N2a^eGFP^.

### 4.2. Quantification Total Levels of GM1

Confluent N2a^eGFP^ and N2a^GPR37-eGFP^ cells were lysed in lysis buffer (150 mM NaCl, 50 mM Tris-Cl, and 1% Triton-X) with 1x protease inhibitors (Roche, Basel, Switzerland). After lysis at 4 °C, lysate was centrifuged for 10 min at 14,000 × *g*; the subsequent supernatant was stored at −80 °C. Protein concentration was determined with BCA Protein Assay (ThermoFisher, Waltham, MA, USA). A total of 5 µg of total protein was mixed with methanol (1:1) and loaded onto nitrocellulose membrane (Bio-Rad, Hercules, CA, USA). The membrane was blocked in 5% nonfat dried milk and incubated for 30 min with CtxB-Alexa 647 (ThermoFisher, Waltham, MA, USA) [44], commonly used for labeling GM1 [45]. ChemiDoc imaging system (Bio-Rad, Hercules, CA, USA) was used for imaging, and intensity was quantified in ImageJ. The average of three technical replicates from three independent experiments was analyzed, and the values were normalized to the fluorescent intensity of N2a^eGFP^.

### 4.3. Quantification of GM1 Expression in Plasma Membrane

1 × 10^4^ N2a^eGFP^ and N2a^GPR37-eGFP^ cells were seeded in an 8-well chamber with a cover slide bottom (Nunc, ThermoFisher, Waltham, MA, USA). Cells were washed in ice-cold HBSS, and CtxB-Alexa 647 in 0.1% BSA was added for 5 min in line with published protocol for labeling the plasma membrane fraction of GM1 [35]. Cells were washed twice in PBS and fixed in 4% PFA for 10 min. Cells were imaged in a Z-stack using Zeiss LSM 880 Airyscan confocal laser scanning microscope using a Plan-Apochromat 20x/0,8 objective. ImageJ was used to merge Z-stack, and the fluorescence intensity at the plasma membrane was quantified in single eGFP-positive cells in order to correctly quantify the entire plasma membrane. In total, 53 cells from three separate experiments were quantified for each condition and normalized to fluorescent intensity of N2a^eGFP^.

### 4.4. Resazurin Assay

2.5 × 10^3^. stable N2a^eGFP^ and N2a^GPR37-eGFP^ cells were seeded in a nontransparent 96-well plate with optical bottom and then left for 24 h. Cells were then treated with GM1 (Enzo Life Sciences, Lausen, Switzerland) or vehicle and incubated for 24 h, after which they were treated with MPP+ (Sigma-Aldrich, St. Louis, MO, USA). After an additional 24 h, resazurin, with a final concentration of 15 µg/mL, was added, and fluorescence intensity was read at 540/590 nm on a Spark 10M plate reader (Tecan, Männerdorf, Switzerland) after 2 h incubation at 37 °C.

For transient transfections, N2a^WT^ cells were seeded in a 384-well plate and transfected with 10 ng eGFP-, GPR37-eGFP, or dopamine receptor 2 (D2R)-TdTomato-plasmid using Lipofectamine 2000. Twenty-four hours after transfection, cells were visually inspected for equal fluorescence expression and treated as above. Forty-eight hours after MPP+ treatment, resuzurin was added, and fluorescence was quantified.

### 4.5. qPCR

N2a^eGFP^ and N2a^GPR37-eGFP^ were seeded in 48-well plates (5 × 10^3^ cells per well) and treated with GM1 at different concentrations. After 48 h, cells were washed in PBS and lysed, and RNA was then extracted with RNeasy Plus Mini Kit (Qiagen, Hilden, Germany). RNA concentration and quality were determined using Spark 10M with a NanoQuant inset or a Nanodrop. iScript cDNA synthesis kit was used for cDNA preparation (Bio-Rad, Hercules, CA, USA). qPCR was performed with 2 ng of cDNA per reaction and Sso Advanced Universal SYBR green super mix (Bio-Rad, Hercules, CA, USA). The following primers were used: RLP19 5′-aatcgccaatgccaactc-3′ and 5′-ggaatggacagtcacagg-3′, GPR37 5′-ccaagacggccaatggactg-3′ and 5′-cggttcgtgctgtttccc-3′, B3GALT4 5′-tgctgtgcagctcatcctg-3′ and gcaccagccaatttgacacagt-3′. The reaction was initially at 95 °C for 30 s and then 40 cycles of 95 °C for 15 s, followed by 60 °C for 30 s. Melt curve analysis was preformed to ensure specificity of amplicons. Ct-values were converted to RNA expression levels using the 2^−^^ΔΔCt^ method.

### 4.6. Signaling

cAMP production was assessed using a Homogeneous Time-Resolved Fluorescence (HTRF) cAMP G_i_ kit (Cisbio, Codolet, France). In short, cAMP-cryptate, a donor, is added to lysed cells to compete with intracellular cAMP to bind a monoclonal-cAMP-d2 antibody, which acts as an acceptor. In case cAMP-cryptate binds to the antibody, the donor transfers energy to the acceptor, commonly called FRET, and a fluorescent signal is detected with a lag time. 6 × 10^4^ cells HEK 293T cells were seeded in 384-well Greiner low volume plates, coated with Poly-D-Lysine (100 µg/mL, Merck Millipore, Burlington, VT, USA) and transfected as explained above. Forty-eight hours after seeding, cells were incubated with 1x stimulation buffer, including 100 µM of Ro20-1724 (Santa Cruz Biotechnology, Dallas, TX, USA), a phosphodiesterase inhibitor, and GM1 or vehicle. After 2 h, cells were stimulated with 5 µM forskolin and incubated for 45 min. cAMP-crypate and monoclonal anti-cAMP-d2 were added, and the TRF signal was measured on Spark 10 M at 320/620 nm and 320/655 nm, respectively. cAMP levels were determined by standard curve interpretation. Data are presented as percentage of forskolin-stimulated eGFP-transfected cells at baseline. On each plate, transfection with D2R was included and stimulated with 10 µM of quinpirole in order to have a positive control.

### 4.7. Statistical Analysis

Comparisons including only one variable and two groups (N2a^GPR37-eGFP^ and N2a^eGFP^) were analyzed using Student’s *t*-test or Mann–Whitney test, depending on normality of distribution. For three groups with one variable, Kruskal–Wallis one-way analysis of variance with Sidak post hoc was used. For two variables, data were analyzed with two-way ANOVA using Tukey’s post hoc test, as noted in the figure legends. GraphPad Prism v. 7.0 was used for statistical analysis. Significance level was set to *p* < 0.05.

## 5. Conclusions

Both GPR37 and GM1 have been implicated in neuroprotection. Here, we show a GPR37-dependent rescue effect of GM1 against a common dopaminergic neurotoxin, MPP+. This effect was seen in both stably and transiently overexpressing N2a cells. In addition, we propose a direct effect of GM1 on GPR37 signaling pathway detected as reduction in constitutive activity of GPR37. This proposed functional interaction should be further evaluated in vivo models to elucidate physiological relevance.

## Figures and Tables

**Figure 1 ijms-22-12859-f001:**
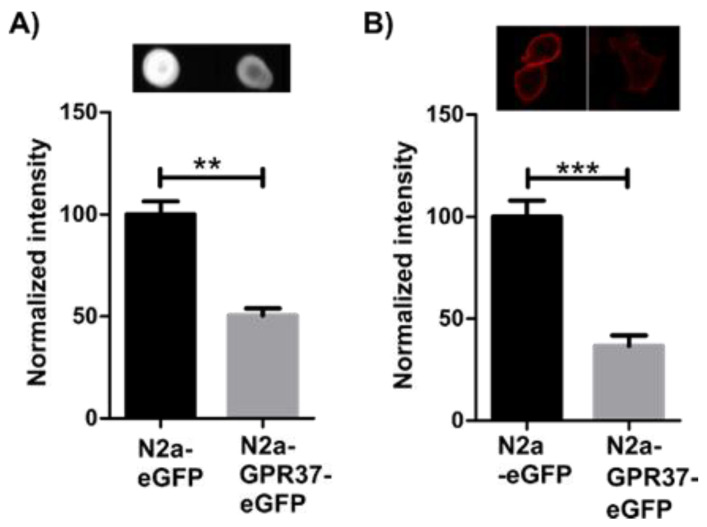
Quantification of total GM1 levels in stable N2a^GPR37-eGFP^ and N2a^eGFP^ detected with Cholera Toxin subunit B (CtxB)-Alexa 647 in dot blot, showing reduced levels of GM1 in N2a^GPR37-eGFP^ compared to N2a^eGFP^. Representative image and data analyzed using *t*-test, n = 3 (**A**). Fluorescent imaging and quantification of plasma membrane fraction of GM1 bound to CtxB-Alexa 647 in red in stable N2a^GPR37-eGFP^ and N2a^eGFP^. A total of 53 cells from three independent experiments was analyzed using non-parametric Mann–Whitney test. (**B**) Data are normalized to fluorescent intensity of N2a^eGF^. Results presented with means and error bars indicate SEM, ** *p* < 0.01, *** *p* < 0.001.

**Figure 2 ijms-22-12859-f002:**
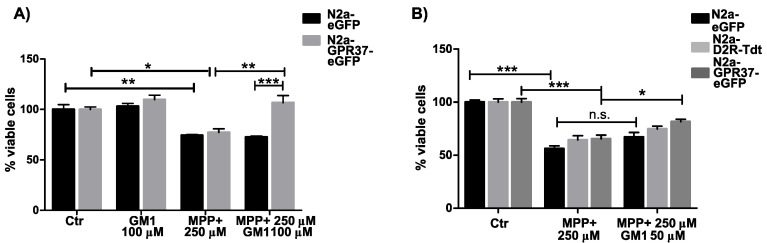
GPR37-specific rescue with 24 h GM1 pretreatment (100 µM) against MPP+ treatment in stable N2a^GPR37-eGFP^ compared to N2a^eGFP^ measured by resazurin assay 24 h after toxin addition. (**A**). Transiently transfected N2a showed the same pattern with GPR37-specific rescue with GM1 treatment (50 µM) 48 h after MPP+ addition. (**B**) Data from three independent experiments. Fluorescence is normalized to the average of control condition. Two-way ANOVA followed by Tukey post hoc test. Error bars indicate SEM. * *p* < 0.05, ** *p* < 0.01, *** *p* < 0.001; n.s., non-significant.

**Figure 3 ijms-22-12859-f003:**
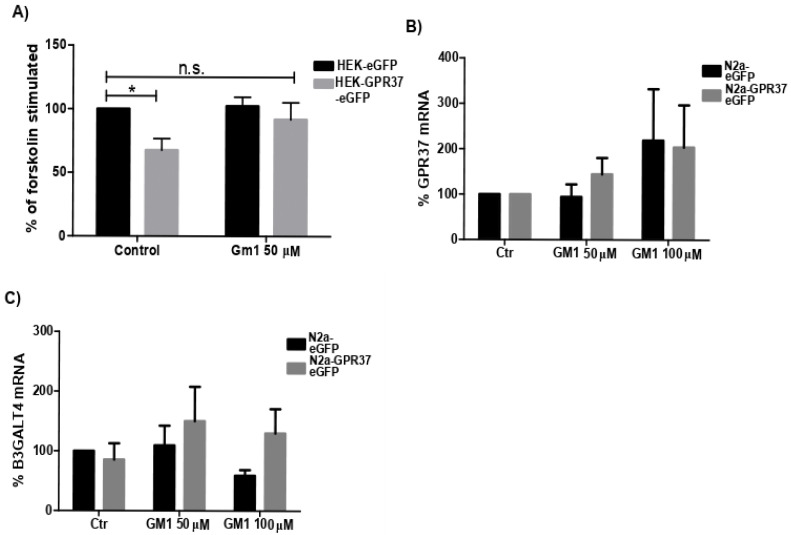
Cellular cAMP level measured by Homogeneous Time-Resolved Fluorescence (HTRF), expressed as percentage of forskolin-stimulated, transiently transfected HEK293T-^eGFP^, were lower in HEK293T-^GPR37-eGFP^ cells under control conditions. The cAMP level was increased in HEK293T-^GPR37-eGFP^ specifically after 2 h of GM1 treatment, indicating shift in GPCR signaling cascade. n = 3 and data analyzed with two-way ANOVA Turkey post hoc test. (**A**) GPR37 RNA expression levels were not significantly changed due to GM1 treatment compared to baseline of each cell type measured by qPCR. n = 3, data analyzed with Kruskal–Wallis test with Sidak post hoc due to non-normal distribution. (**B**) qPCR of B3GALT4 RNA showed that expression levels were not reduced in N2a^GPR37-eGFP^ at baseline and was not regulated by GM1 treatment. n = 3, data analyzed with two-way ANOVA Turkey post hoc test. (**C**) Error bars indicate SEM * *p* < 0.05. n.s., non-significant.

## Data Availability

Data is available upon request.

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
