# Peer review of "GM1 Is Cytoprotective in GPR37-Expressing Cells and Downregulates Signaling"

_ijms, 2021, doi:10.3390/ijms222312859_

Round 1

Reviewer 1 Report

In this manuscript, Hertz, E et al, address the functional consequences of known interaction between brain ganglioside GM1 and G protein-coupled receptor 37 (GPCR-37). Both GM1 and GPCR-37 has tremendous clinical significance in connection to neurodegenerative diseases and have been extensively studied. The neuroprotective effect of GM1 is well known whereas GPCR-37 has shown to have both neurotoxic and neuroprotective effects. The present study suggests that the interaction of GM1 with GPCR-37 on the plasma membrane possibly release the inhibitory effect of GPCR-37 on cAMP levels activating the signaling pathway downstream of GPCR-37.

The manuscript is well written. The text provides sufficient background information, the objective of the study is clearly stated and the language and style ensure productive reading.   The experiments are designed and executed well and data presented are of sufficient quantity and quality to justify publication as a short communication. However, I have the following concern.

The authors have used cell viability assay to show the neuroprotective effect of GM1 following MPP+ treatment (Result 2.2). However, the results are less convincing in terms of showing the specificity of GM1 in rescuing the cells (Fig 2B) as the two other tested constructs N2a-eGFP and N2a-D2R Tdt show similar effects. However, I noticed that the authors used a maximum GM1 concentration of 50μM in this experiment, while they have used 100μM GM1 in another experiment and the rescuing effect of GM1 is more profound (Fig 2A).  It will be helpful to see how the other two tested cell lines (N2a-eGFP and N2s-D2R Tdt, respond to 100μM GM1. If this has been already done, please discuss it in the text. Otherwise, I suggest repeating the experiment with higher doses of GM1 (at least 100μM). This will be an important piece of information for improving the quality of data.  

Reviewer 2 Report

The authors present an interesting concept, where the ganglioside GM1 offers a cytoprotective affect mediated at least in part through GPR37.  While this work is interesting based of of how little is known about the mechanism of GM1 protection and the function of GPR37, this reviewer finds it extremely difficult to draw any significant conclusions from the data as presented.  Serious concerns about statistical methods, limited information about the methods employed, and severely deficient figure legends result in very low enthusiasm about this manuscript in its current form.  The following major comments reflect this inability to draw any conclusions from the data presented, and thus an inability to evaluate the authors’ interpretation of the data.  As there is no way to evaluate the data conclusively as presented, and no indication that the panels represent more than 1 independent experiment, I must suggest that this manuscript be rejected at this time. 

Method of GM1 quantification:  The indicated citation detailing this method does not describe the method used in these experiments.  Please reference the method itself, as the binding of cholera toxin subunit B to GM1 is widely accepted and would undoubtably be supported in any work outlining the actual method employed. 

Statistical methods:  Only figure 1 displays data comparing one variable and two groups, however there are two options listed in the statistical analysis of this data based on distribution (Figure 1).  Does this indicate that there was different distribution characteristics in data between panels A and B?  This should be clarified in the figure legend itself.

Figures- the figure legends are insufficient to describe the associated experiments adequately.

For example:

Figure 1:  There is no indication of n, resulting in an inability to determine if SEM is the appropriate manner to describe these means.  As SEM indicates uncertainty within the mean, vs. SD which indicates variability of the mean, it is often more appropriate to use SD rather than SEM.  Additionally, there is no mention of reproducibility in independent replicate experiments, limiting this reviewer’s ability to draw a conclusion about the robust nature of these measurements.  Furthermore, there is no indication of the statistical analysis used to generate the p values.  Finally, what is the intensity normalized to?  This is not listed in the figure legend or the methods section.

Figure 2:  The legend in panel B is not explained in the figure’s text.  Additionally, there should be a western showing the level of overexpression in the transiently transfected N2a.  Also, there does not appear to be any figure showing the overexpression in the stably transfected cell lines.  There are identical concerns about the n and independent experiments in this figure.  Moreover, there is no indication that there were significant amounts of cell death induced by MPP+ compared to control cells, limiting the usefulness of the results showing differences in cell death between the cells treated with GM1 and MPP+ and those treated with MPP+ alone.  There is no indication of this difference in the supplementary figures associated with this figure either, suggesting that there is limited impact of MPP+ treatment and while GM1 corrects the killing of cells by MPP+ that killing itself is not significantly observed in this model.  As the post hoc analysis (Tukey) would have compared all the means to each other mean, this data would be present in the original analysis and thus should be included.

Figure 3:  There are identical concerns about the n and independent experiments in this figure.  B- the magnitude of the error bars, which are indicated to be reflective of SEM, are concerning.  It is hard to draw any conclusion from this data as it appears that reproducibility is a major problem.  Additionally, the methods section indicates that panels showing three groups would have been analyzed using Kruskal-Wallis with Sidak, however the legend indicates that a 2 way ANOVA and Tukey were used.

These fundamental design and presentation issues will need to be addressed before any evaluation of the actual content of this work can be evaluated.

Round 2

Reviewer 2 Report

I would like to thank the authors for their careful consideration of my concerns, of which all have been adequately addressed.